# Forecasting Economic Recession through Share Price in the Logistics Industry with Artificial Intelligence (AI)

**YM Tang** [1], **Ka-Yin Chau** [2], **Wenqiang Li** [1,*] **and TW Wan** [1]

1   Department of Industrial and Systems Engineering, The Hong Kong Polytechnic University, Hung Hom, Hong Kong, China; mfymtang@polyu.edu.hk (Y.T.); 16067099d@connect.polyu.hk (T.W.)
2   Faculty of International Tourism and Management, City University of Macau, Macau, China; gavinchau@cityu.mo
*   Correspondence: wenqiang.li@connect.polyu.hk

**Abstract:** Time series forecasting technology and related applications for stock price forecasting are gradually receiving attention. These approaches can be a great help in making decisions based on historical information to predict possible future situations. This research aims at establishing forecasting models with deep learning technology for share price prediction in the logistics industry. The historical share price data of five logistics companies in Hong Kong were collected and trained with various time series forecasting algorithms. Based on the Mean Absolute Percentage Error (MAPE) results, we adopted Long Short-Term Memory (LSTM) as the methodology to further predict share price. The proposed LSTM model was trained with different hyperparameters and validated by the Root Mean Square Error (RMSE). In this study, we found various optimal parameters for the proposed LSTM model for six different logistics stocks in Hong Kong, and the best RMSE result was 0.43%. Finally, we can forecast economic recessions through the prediction of the stocks, using the LSTM model.

**Keywords:** artificial intelligence; share price; stock quote; long short-term memory; logistics; big data

---

## 1. Introduction

In 2019, over 2.1 million Hong Kong stock investors contributed 10.3% of the cash marketing trading value, which was valued at HK$2.6 trillion [1]. Given the vast amount of stock transactions, importantly, there was a distinct decline of transaction value in the overall Hong Kong investment market. Especially for local retailer investors, the total market turnover amount dropped from HK$240 billion in 2016 to HK$150 billion in 2018. The decline in turnover could be attributed to some unexpected events. For example, Hong Kong politics becoming unstable, trade war leading to conflict between the US and China, China controlling capital outflows, the US presidential election, and Brexit [2].

To seek investment opportunities, local retail investors need to spend a great deal of time, while relatively wealthy and independent investors usually seek help from professional consultation services [3]. For typical retail investors, high consultation costs are impossible and impractical. Therefore, local retail investors are under tremendous pressure to make, buy, or sell decisions since they have to figure out the current situation of the investment market and determine the trend of stock prices.

Without assistance from qualitative and quantitative analyses or models [4], people quickly become irrational. When a decision falters due to cognition bias or emotional sentiment, an unnecessary loss is

incurred. Although retail investors are cautious enough, most of them do not have sufficient data or are not skillful enough to process the historical stock prices to make a correct judgment. In contrast, institutional investors can apply various models with advanced technologies to avoid investment traps. Retail investors find themselves falling behind in investment marketing due to the restriction on the usage of these technologies. If quantitative and data-driven models cannot be accessed, retail investors typically use some distinctive indicators to evaluate the stock market. For instance, linear regression [5] and Exponential Moving Average (EMA) [6] are standard tools to predict stock prices. Twenty-day EMA and 50-day EMA are two critical indicators. When the 20-day EMA curve is below the 50-day EMA curve, it means the stock price is trending downwards, and vice versa. This situation is shown in Figure 1. Retail investors probably use another tool, linear regression, by linking the most significant and smallest candlesticks to predict the stock price.

Forecasting algorithms become ubiquitous, which inspires the application of stock prediction. These algorithms may become potential methods to discover the hidden modes behind the stock trend. The resulting data provide a practical insight for retail investors who need to make buy-or-sell decisions in a determined period.

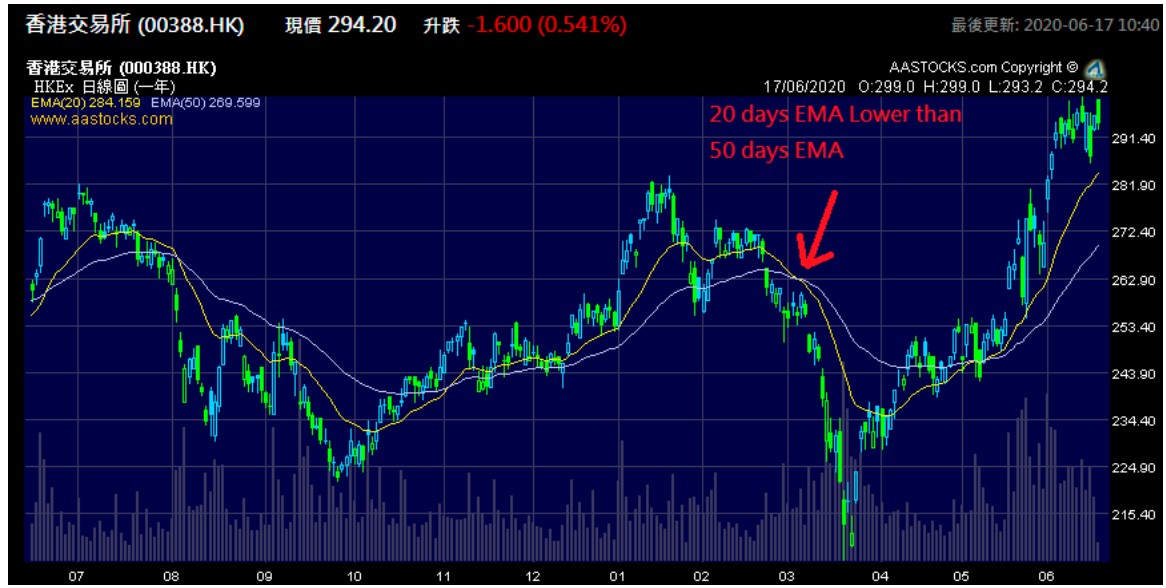

**Figure 1.** The 20-day Exponential Moving Average (EMA) of the share price, which is lower than the 50-day EMA of an HK listed stock [7].

Despite the fact that many different statistical approaches have been adopted to forecast share prices [8] due to the share prices and their trends being sets of time series data, the non-linear and non-stationary characteristics of these data make the forecasting challenging and stressful. The share price can be considered as supply and demand trading, which is affected by several factors, such as its political and economic situations, and financial crises. The influences of these events and crises are all reflected in the share prices in numerous industries, such as energy, financial, transportation and logistics, health care, industrial, etc. The events and crises impact not only on the share price but also on the economic downside, financial risk, and economic recession. The share price is usually anticipated by business revenues and reflects the slowing down of business activities [9].

There have been lots of related studies conducted on economic recession prediction. In 2001, Qi employed a novel neural network (NN) to predict the probability of economic recession based on the leading indicators, such as the Stock and Watson index, S&P500 index, and so on [10]. Another closely related study adopted a probit framework to predict economic recessions by the yield curve and stock market liquidity [11]. However, with the development of machine learning and artificial intelligence, more and more economic recession models have been proposed with these latest technologies [12].

Due to the advance in artificial intelligence (AI) technology development nowadays, AI tools are now extensively used in forecasting, prediction, and other intelligence applications [13], including stock market investment [14]. Heiberger proposed a naïve Bayes model to predict the economic growth based on stock network data [15], and similarly, in this paper, we propose a Long Short-Term Memory (LSTM) economic recession prediction model based on the logistics share price. Among different kinds of industries and business sectors, logistics companies are believed to provide the important indicators to anticipate the economic growth and financial performance of companies [16,17].

In order to alleviate the risk of investment for retail investors, we have developed a share price prediction model of a logistics company in Hong Kong. Investors can anticipate the economic recession based on the results of the model, and further avoid potential investment risks in advance. Therefore, in this project, we aim to adopt advanced artificial intelligence (AI) technology to forecast the share price in the logistics industry. Through such forecasting, we intend to foresee the possibility of a financial recession. As Hong Kong is one of the leading international financial centers, the logistics companies are listed in the Hong Kong exchange market. In this study, we intend to verify the algorithm's capability to forecast stock prices in Hong Kong. We compare the performance of different algorithms to forecast stock trends and optimize the model to achieve a more accurate result for stock quote prediction.

This paper consists of five parts with a standard structure. Section 1 illustrates the motivation of the study and the in-depth literature review on the related works. A comparative analysis is conducted to validate the result. Section 2 introduces the historical stock price dataset and explains the methods and validation algorithms used in this study. Section 3 presents and interprets the results. Finally, Sections 4 and 5 discuss the findings, the limitations, and conclusions of the study.

## 2. Materials and Methods

This investigation is implemented in several stages. In the first stage, the historical stock price data are collected from the public platform Yahoo Finance. The data collection period of the stock price is predefined to reduce the amount of data. The second stage is data pre-processing, where any missing data and null cells are removed since such incorrect data will lead to inaccurate training of the model and predicted results. In the third stage, the captured data are split into the training and testing datasets so that a suitable amount of data can be used to train each stock model. In the fourth stage, the data are input to the selected model to predict the stock price after the completion of model training based on the historical stock price. In this article, five algorithms are implemented for stock price forecasting to make a comparison—the forecasting algorithms, the Moving Average (MA), K-Nearest Neighbor (KNN), Autoregressive Integrated Moving Average (ARIMA), Prophet, and LSTM. The Root Mean Square Error (RMSE) is used to quantify the performance in the fifth stage. In this stage, we review the models' effectiveness and model accuracy. If the result proves the LSTM model outperforms other models, we implement further optimization with different hyperparameters of the LSTM model.

### 2.1. Data Collection

In this study, we adopt Yahoo Finance to collect the stock quote data. Moreover, we utilize the adjusted closing price as the input of various forecasting algorithms to identify the price trend and pattern. Although we only adopt one industry for investigation, there are still many different features and distinct price trends for each stock. Thus, each stock model is built separately for each algorithm. Hong Kong is one of the leading international financial centers; therefore, we adopt the listed stocks in Hong Kong for the logistics industry in this study. In order to select suitable stocks, some criteria are considered, such as the characteristics of the stock, the volatility of the stock price, company size, popularity, and related companies. The selected stocks in the logistics industry from Yahoo Finance are listed below:

- China Merchants Port Holdings Company Limited (0144.HK)
- Li & Fung Limited (0494.HK)
- Sinotrans Limited (0598.HK)
- Kerry Logistics Network Limited (0636.HK)
- COSCO SHIPPING Ports Limited (1199.HK)
- China COSCO Holdings Company Limited (1919.HK)

Specific stock data are captured from the Yahoo Finance library automatically through a Python API. All databases have a unique stock code for each stock. The data time is from 1 January 2015 to 31 December 2019. The data range includes open price, close price, low price, high price, adjusted close price, and volume. After inputting the stock code and specified data period, the historical stock price data are read from Yahoo Finance. There are 1232 rows of data for each stock. The collected data are then saved in the CSV (comma-separated values) format and named according to their stock codes for further analysis. The date is set as an index so that the time series stock price data can be formed. Figure 2 is the visualized graph for the stock adjusted closing price.

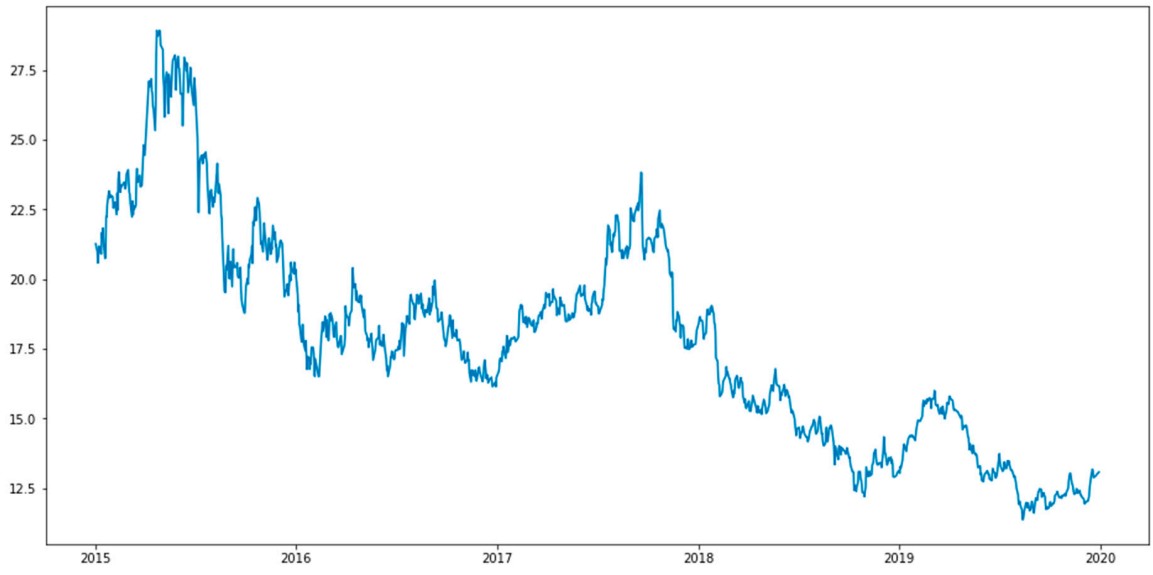

**Figure 2.** Visualizing stock price time series data.

*2.2. Data Pre-Processing*

Data pre-processing is a significant step before building a model. The steps in data pre-processing include cleaning, normalization, the transformation of raw data, and feature extraction. Performing data pre-processing can remove noise from the target set and solve the problem of missing values. The data are divided into training, validation, and testing sets. In this stage, we mainly focus on the training dataset and testing dataset only. Therefore, we divide the original data into two sets of data, the training set and the test set, by ratio. In order to test the accuracy of different ratios, we set the ratios of the training set to the testing set as 90%:10%, 80%:20%, 70%:30%, and 60%:40%, respectively. The training set is used to train the model, and the testing set is used to assess the performance of the model.

*2.3. Forcasting Algorithms*

2.3.1. Moving Average (MA)

MA is one of the simplest and most common time series forecasting methods. MA is also called simple MA. Within the sampling period (daily, weekly, or monthly), stock prices from the current period and the previous periods are summed and divided by the number of dates in the periods to

get a simple average. Then, moving to the next period, a new simple average is calculated. However, when calculating the new simple average, the stock price from the latest period is added; and at the same time, the stock price from the earliest period is removed. The equation below is an example of the calculation of the simple MA of the stock price in days, while the summation stock closing prices from each day within the sampling period are divided by the number of sampling days.

$$Simple\ Moving\ Average = \frac{Summation\ of\ stocks\ closing\ price\ within\ sample\ period}{Number\ of\ data\ in\ sample\ period}, \tag{1}$$

Simple MA can also be expressed as statistic equation:

$$MA_t(T) = \left(\frac{1}{T}\right)\sum P_{t-1} = \left(\frac{1}{T}\right) + (P_t + P_{t-1} + \cdots + P_{t-T+1}), \tag{2}$$

In the above equation, $t$ is the $t$th period and $T$ is the sum of periods.

The predicted stock price is the average of the previous historical data. In this project, the MA method uses the most recent data to calculate the predicted stock price. In other words, for each subsequent new time, the oldest stock price is deleted from the set, and the last stock price is added to predict the stock price.

### 2.3.2. K-nearest Neighbor (KNN)

K-nearest neighbor (KNN) [18] is a machine learning algorithm under supervised learning. Regression from the KNN algorithm, mainly for continuous variable data, is the chosen method in this project. Similarly, classification is the first step. Then the average value of the category is assigned to the predicted value. The value of the predicted point is obtained by averaging the distance between the K points. For the logistic share price prediction, we adopt nine neighbors, and the distance as the Manhattan distance. Besides, all the points in the KNN model are weighted equally.

### 2.3.3. Autoregressive Integrated Moving Average (ARIMA)

The Autoregressive Integrated Moving Average (ARIMA) model [19] is derived from the Autoregressive (AR), Moving Average (MA), and Autoregressive Moving Average (ARMA) models. It was proposed by Box and Jenkins in 1970. There are three critical parameters in ARIMA: $p$ (the past value used to predict the next value), $q$ (past prediction error used to predict future values), and $d$ (order of differencing). ARIMA parameter optimization requires much time. Therefore, ARIMA can automatically select the best combination ($p$, $q$, $d$) with the smallest error. The Akaike information criterion (AIC) and Bayesian information criterion (BIC) are used to calculate the score for each parameter combination. AIC and BIC are accurate scores to determine the best combination, but the amount of calculation is enormous. Additionally, seasonality is not considered and has to be calculated individually. The best parameter combination is selected with the least AIC and BIC. The training set and testing set are fitted to the model to make predictions. ARIMA considers past values and residuals as well as stationary problems at the same time. The ARIMA model can be used to analyze a non-stationary time series by turning the originally non-stationary time series into a stationary time series after d differencing. The differential autoregressive moving average model is written as ARIMA ($p$, $d$, $q$):

$$\left(1 - \sum\nolimits_{i=1}^{p} \varnothing_i L^i\right)(1-L)^d X_t = \left(1 + \sum\nolimits_{j=1}^{q} \theta_j L^j\right)\varepsilon_t, \tag{3}$$

### 2.3.4. Prophet

In 2017, Facebook introduced Prophet [20], a time series prediction framework currently supporting the R language and Python. Prophet is an open-source library based on a decomposable model (trend + season + holiday), and fully integrates knowledge of the business background and statistical knowledge. At the same, Prophet also supports customization of the effects of seasons and holidays. Prophet deals

with some outliers in the time series and some missing values. This algorithm is based on time series decomposition and machine learning fittings.

The whole process of Prophet is divided into four parts: modeling, forecast evaluation, surface problems, and visually inspect forecasts. Modeling means establishing a time series model, then selecting a suitable model based on the background of the forecast problem. Forecast evaluation refers to model evaluation. According to the model, simulation is conducted on the historical price. When the parameters of the model are uncertain, we can make multiple attempts and evaluate which model is more suitable according to the corresponding simulation results. Surface problems means presenting the problems. Although numerous parameters have been tried, the overall performance of the model is still not ideal. The potential causes of larger errors can be presented to the analyst. The visually inspect forecasting step is used to visualize the predicted results. When the problem is fed back to the analyst, the analyst considers whether to further adjust and build the model. Prophet is the process used which combines background knowledge and statistical analysis. It dramatically increases the scope of the model and improves the model accuracy.

### 2.3.5. Long Short-Term Memory (LSTM)

The Long Short-Term Memory (LSTM) model [21] is an evolution of the Recurrent Neural Network (RNN) and has also been widely used in the prediction of stock prices and many derivative financial commodities. LSTM is very powerful in solving sequence prediction problems because it can store previous information, which is essential to predict the stock price's future trend. When LSTM processes data from a time series, its predicted errors or results are superior to the general Recurrent Neural Network model, since LSTM solves the problems of gradient disappearance and gradient explosion during long sequence training. Figure 3 shows the structure of the LSTM layer. $c^t$ means cell state, $h^t$ means hidden state, $x^t$ is the input, and $y^t$ is the output. Among them, $c^t$ changes slowly and $c^t$ equals $c^{t-1}$ adding the output value from previous layer. However, the difference between $h^t$ and $h^{t-1}$ may be very large, and $h^t$ also changes fast.

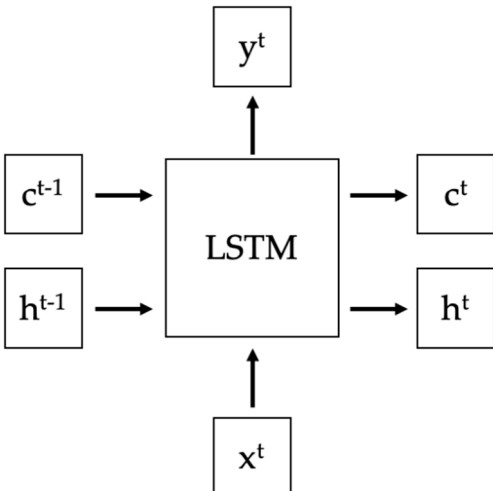

**Figure 3.** Structure of the Long Short-Term Memory (LSTM) layer.

Figure 4 shows the overall process of the LSTM model training and validation. The initial stock price data are normalized, ranging from 0 to 1. In the meantime, the dataset is randomly divided into training, validation, and testing datasets, which take 80%, 10%, and 10%, respectively. The training set is initially created with time steps of 60 and converted into an array with three dimensions. When the time step is set to be 60, each step represents a feature. In order to prevent overfitting, the LSTM layer and dropout layer are added. The dropout layer is defined as 0.3, which means that 30% of each layer is deleted randomly. After the normalized training data go through the LSTM and dropout layer,

they runs into the dense layer. In the meantime, the data shape changes from (none, 19, 4) to (none, 4). Besides, the model uses Adam as the optimizer for compiling, and the loss function is the mean square error (MSE) to calculate the error and improve the accuracy through backpropagation. The testing dataset goes through the trained model with the optimal parameters, and the output is the predicted stock price. We calculate the RMSE based on the real stock price and predicted stock price to assess the proposed model's accuracy.

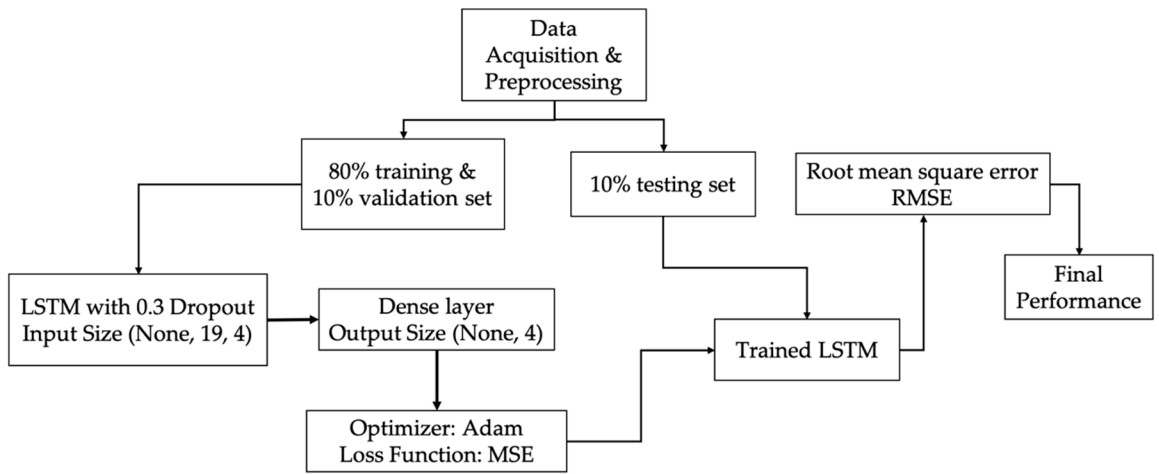

**Figure 4.** Process of the proposed LSTM.

After the LSTM model is trained with historical data and customized parameters, the error between the predicted valid price and actual valid price is calculated as the Mean Square Error (MSE), whereas the error between predicted test price and actual test price is calculated as the Root Mean Square Error (RMSE). The predicted stock price is visualized Figure 5:

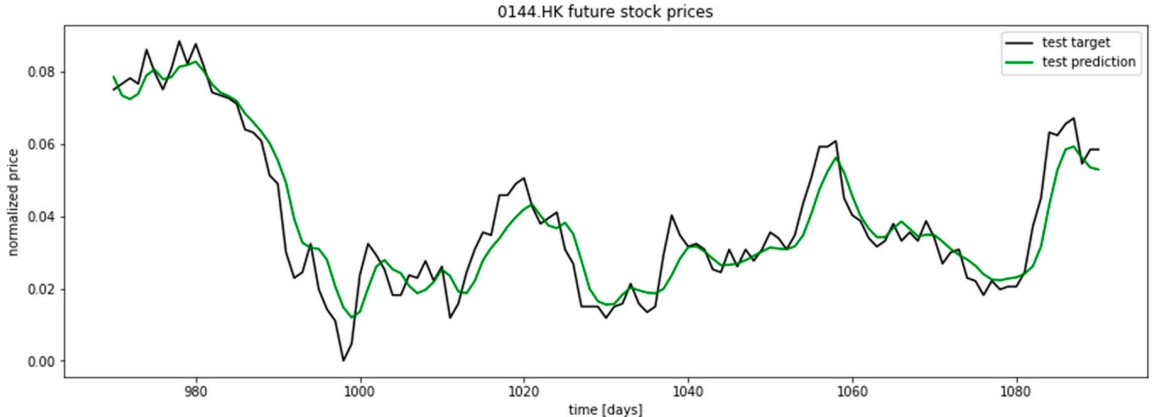

**Figure 5.** Future stock price prediction of 0144.HK based on LSTM.

*2.4. Evaluation Metrics*

After testing the stock price with different sizes of training data and different algorithms, we summarize the result from the same ratio of training and testing set and show the results in Figure 6. Each colored line represents the result of each test. Furthermore, we calculated the Root Mean Square Error (RMSE) and Mean Absolute Percentage Error (MAPE) to show the error in each test.

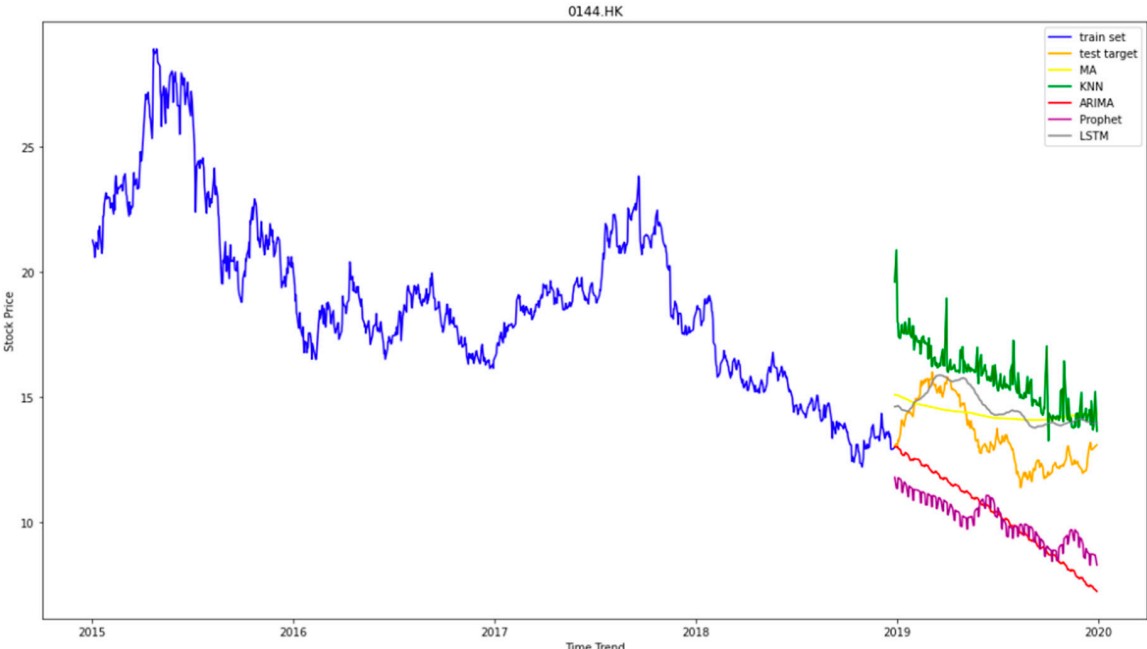

**Figure 6.** Summarization of 0144.HK stock prediction of all models.

The Root Mean Square Error (RMSE) is the average square root of the error between the predicted value and the actual value. The RMSE is also known as the standard error. The standard deviation is used to set the degree of dispersion of a set of numbers, and the root mean square error is used to observe the deviation between the observed value and the real value. It is very sensitive to large or small errors in a group of measurements—the smaller the value of the RMSE, the higher the accuracy of the model. Therefore, the RMSE can well reflect the precision of the measurement and can be used as a standard for evaluating accuracy within the same model. The equation of RMSE is as follows:

$$RMSE = \sqrt{\frac{1}{n}\sum\nolimits_{i=1}^{n}(\hat{y}_i - y_i)^2}, \tag{4}$$

The Mean Absolute Percentage Error (MAPE) is one of the most commonly used indicators to evaluate prediction accuracy. The MAPE is the sum of each absolute error divided by the actual value. Namely, it is the average of the percentage error. Sometimes, the MAPE is a very particular performance evaluation index. It can be indicated from the formula that the MAPE divides each error value by the actual value, so there is skew; if the actual value at a particular time is deficient, and the error is significant, it has a high impact on the value of the MAPE. Therefore, it can well reflect the precision between models—the smaller the value of the MAPE, the higher the accuracy of that model. The equation for the MAPE is shown as follows:

$$MAPE = \frac{100\%}{n}\sum\nolimits_{i=1}^{n}\left|\frac{\hat{y}_i - y_i}{y_i}\right|, \tag{5}$$

## 3. Results

### 3.1. Performance of Different Forecasting Approaches

In this section, we mainly focus on the MAPE values by applying the same stock data to different models. Through analyzing the MAPE value, the accuracy of model prediction can be determined. The results are shown in Table 1 and Figure 7:

**Table 1.** Comparison table of model performance in the MAPE.

| | Stock Code | | | | | |
| --- | --- | --- | --- | --- | --- | --- |
| | **0144** | **0494** | **0598** | **0636** | **1199** | **1919** |
| MA | 10.44% | 34.69% | 19.72% | 4.78% | 7.69% | 10.05% |
| KNN | 13.55% | 120.09% | 26.15% | 13.9% | 11.27% | 18.56% |
| ARIMA | 4.42% | 8.57% | 8.51% | 6.34% | 12.25% | 4.77% |
| PROPHET | 10.06% | 10.84% | 6.03% | 5.28% | 10.7% | 7.55% |
| **LSTM** | **3.27%** | **6.97%** | **3.16%** | **3.81%** | **3.63%** | **4.24%** |

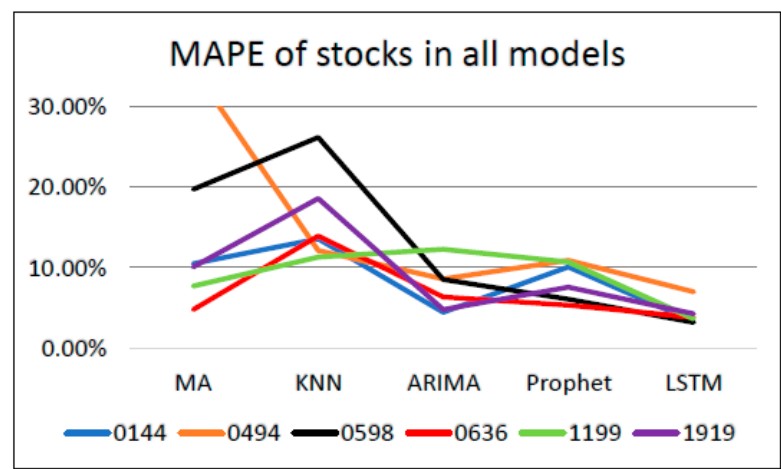

**Figure 7.** Comparison graph of model performance in the Mean Absolute Percentage Error (MAPE).

We compare the model performance within the same stock in Table 1, and the MAPE value is analyzed in the same column. Since the value of the MAPE indicates the precision of the model, the lower the value of the MAPE, the better the performance of the model. When the model shows a comparatively lower MAPE value than others for one stock, that model performs relatively more accurately than others. LSTM performs the best in all six stocks with the smallest MAPE value in each column. For instance, for stock 0144, the MAPE value of LSTM is 3.27%, which is smaller than the MAPE values of MA (10.44%), KNN (13.55%), ARIMA (4.42%), and Prophet (10.06%), respectively. Moreover, Figure 7 also shows the comparison of the model performance. As labeled in the graph, the difference in the colors stands for different stocks, for example, the blue line means the stock 0144. The MAPE value of LSTM is always the lowest point in Figure 7 for all six stocks. Overall, LSTM outperforms the other models.

*3.2. Optimizing the Long-Short Term Memory (LSTM) Model*

After proving that the LSTM model predicts better than the other models, we further improve the LSTM model's accuracy with various adjustments of the input hyperparameters. The tables below show the result of cross-analysis by classifying the batch size, the number of neurons, and the learning rate.

Table 2 shows that, when the batch size is 20, the RMSE value in the validation and testing periods are the smallest. Therefore, batch size = 20 is the best parameter, followed by batch size = 40, the second-best parameter in the actual prediction of stock 0144.HK. Table 3 indicates that, when the number of neurons is 300, the RMSEs in the validation and testing periods are the smallest. Therefore, number of neurons = 300 is the best parameter in the actual prediction of stock 0144.HK. Moreover, for Table 4, it is found that when the learning rate is 0.5, the RMSE values in the validation and testing periods are the smallest. Therefore, learning rate = 0.5 is the best parameter in the actual prediction of stock 0144.HK.

**Table 2.** Root Mean Square Error (RMSE) values of stock 0144.HK based on different batch sizes.

| Batch Size | Validation Period RMSE | Testing Period RMSE |
|:---:|:---:|:---:|
| **20** | **0.012062658** | **0.009353333** |
| 40 | 0.012075362 | 0.009385105 |
| 60 | 0.012478094 | 0.010352536 |
| 80 | 0.014839811 | 0.011709842 |
| 100 | 0.01662702 | 0.012053032 |

**Table 3.** RMSE values of stock 0144.HK based on different numbers of neurons.

| Number of Neurons | Validation Period RMSE | Testing Period RMSE |
|:---:|:---:|:---:|
| 100 | 0.014442632 | 0.012221126 |
| 200 | 0.01282243 | 0.010681524 |
| **300** | **0.011365698** | **0.008810548** |
| 400 | 0.013806797 | 0.010058999 |
| 500 | 0.015266349 | 0.010324258 |

**Table 4.** RMSE values of stock 0144.HK based on different learning rates.

| Learning Rate | Validation Period RMSE | Testing Period RMSE |
|:---:|:---:|:---:|
| 0.1 | 0.01339052 | 0.010460981 |
| 0.2 | 0.013380858 | 0.010835873 |
| 0.3 | 0.01394538 | 0.0105911 |
| 0.4 | 0.013531418 | 0.010556251 |
| **0.5** | **0.013234768** | **0.010409643** |

The final results of the best combination of parameters are shown in Table 5. The results show that the optimal value of batch size, number of neurons, and learning rate are distinct for every stock. Since the price trends and patterns are different for each stock, the batch size, the number of neurons, and the learning rate should differ.

**Table 5.** Best hyperparameters for all stocks.

| Stock Code | Batch Size | Number of Neurons | Learning Rate |
|:---:|:---:|:---:|:---:|
| 0144.HK | 20 | 300 | 0.5 |
| 0316.HK | 100 | 100 | 0.2 |
| 0494.HK | 40 | 500 | 0.2 |
| 0598.HK | 40 | 300 | 0.3 |
| 0636.HK | 60 | 200 | 0.1 |
| 1199.HK | 20 | 300 | 0.3 |
| 1919.HK | 40 | 400 | 0.5 |

## 4. Discussion

In the development of the modern market economy, the relationship between economics and finance is inseparable. The development of the stock market has become an indispensable part of economic growth. The stock market is one of the essential elements of the financial market. It not only holds the market capital flow but is also responsible for the rational adjustment and allocation of resources and the critical function of controlling financial risks. Therefore, the stock market has a vital position in economic development. As an important indicator for measuring stocks in the market, the stock price reflects the overall level of the listed companies in the stock market. The price level and degree of volatility affect the stock market's investment atmosphere and influence the decision-making direction of investors.

The findings suggest that LSTM comparatively outperforms other models in predicting the future stock price accurately. Besides, the optimization of LSTM is studied in this research as well. A variety of model evaluations to find the most suitable prediction model for each stock can help investors and managers understand the trend of stock price development, and then help both parties make correct decisions and appropriately avoid risks.

From a macroeconomic point of view, the effectiveness of logistics has a more significant impact on the economy. A convenient, efficient, and fast logistics system will bring a company's products and services closer to customers, make inventories more reasonable, and lower the cost of goods circulation, thereby improving the quality and efficiency of macroeconomic operations and promoting the transformation of the national economic growth mode. Therefore, accelerating the development of logistics will surely become a new growth point in the economy. The share price of logistics companies is positively related to the development of the logistics industry, and the development will accelerate the growth of the economy. Therefore, the share price in the logistic industry is positively related to economy growth. The decrease in stock prices in logistics may anticipate a recession in the economy. However, the precise logistic stock price prediction is just the beginning of economic recession study, and the model can be further improved by adding more input factors. In this study, the logistics share price is the only indicator of the recession prediction model, which may lead to one-sided results. Therefore, for the next step, an LSTM with multiple variables can be the future study direction. The input of the model can cover not only the stock price in the logistics industry, but also other indicators, such as yield curve, stock network, Stock and Watson index, S&P500 index, and so on. Despite the stock price in logistic industry reflecting the change in the economic growth to a certain degree, due to the complexity and uncertainty of the economic market, it is still hard to predict economic recession with a single factor.

## 5. Conclusions

The prediction of the movement of the stock market involves data from various perspectives which are the possible influencing factors. Predicting stock prices from big data analysis is not simple since it is hard to obtain complete information from the relevant area for analysis. Therefore, this project mainly focuses on analyzing the stock price as the only factor, proposing an approach to select the best one out of five forecasting models (MA, KNN, ARIMA, Prophet, and LSTM) by comparing the MAPE value of different models. There are six stocks from the logistics industry, which possess 1232 records of data individually within the period from 1 January 2015 to 31 December 2019. After testing with different sizes of training sets, the performance of the five models is also evaluated. No matter what kind of training ratio is obtained, the calculated MAPE is the smallest in LSTM. Overall, LSTM shows a more stable and accurate prediction than other models. The optimal result of our model achieves 0.43% of the RMSE score. Moreover, we can further forecast economic recession by the logistics share price prediction with LSTM based on the relationship between logistics and economic growth.

**Author Contributions:** Conceptualization, Y.T.; Methodology, Y.T. and T.W.; Software, T.W.; Formal analysis, T.W.; Investigation, Y.T. and T.W.; Data curation, Y.T. and T.W.; Writing—original draft preparation, Y.T., K.-Y.C., W.L. and T.W.; Writing—review and editing, Y.T., K.-Y.C., and W.L.; Supervision, Y.T.; Project administration, K.-Y.C. and W.L. All authors have read and agreed to the published version of the manuscript.

**Funding:** This research received no external funding.

**Conflicts of Interest:** The authors declare no conflict of interest.

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
