# Peer review of "Forecasting Economic Recession through Share Price in the Logistics Industry with Artificial Intelligence (AI)"

_computation, doi:10.3390/computation8030070_

Round 1

Reviewer 1 Report

Forecasting Economic Recession Through Share Price in Logistics Industries with Artificial Intelligence (AI)

This paper proposes a long short-term memory unit (LSTM) forecasting model for share price prediction in logistic industry. The proposed model is validated by RMSE. The paper is in very good form; however, I have some comments that must be incorporated before publication.

  1. The abstract requires major revisions to reflect the key ideas of this paper as well as the details of the proposed approach. Also, emphasis on the goals of this article. Please also indicate your findings in percentage.
  2. The motivation should be detailed. The reviewer is unable to catch it.
  3. The first section should include four parts: motivation, literature survey, contributions, and the organization of the manuscript.
  4. Based on table 1, why LSTM is performing better? Please emphasis on this fact.
  5. Please add some references from 2020. The reviewer is unable to find.
  6. Please improve conclusions by adding findings in percentage.

Reviewer 2 Report

The study addresses an interesting topic regarding the use of AI in forecasting Economic Recession. However, it is limited to testing models consecrated in the literature, in order to identify which is more efficient in a given perimeter.
In order to have scientific relevance, the study must be completed with an in-depth review of the literature, identifying similar studies conducted in different areas. In this way a comparative analysis can be performed which will ultimately lead to a reasonable validation of the results.
It is also recommended to involve a phenomenon (temporal or economic) that differentiates the results.

Round 2

Reviewer 2 Report

I agree with the changes made.

Best regards